# Chronic Traumatic Encephalopathy as the Course of Alzheimer’s Disease

**DOI:** 10.3390/ijms25094639

**Published:** 2024-04-24

**Authors:** Magdalena Pszczołowska, Kamil Walczak, Weronika Miśków, Katarzyna Antosz, Joanna Batko, Donata Kurpas, Jerzy Leszek

**Affiliations:** 1Faculty of Medicine, Wroclaw Medical University, Ludwika Pasteura 1, 50-367 Wrocław, Poland; magdalena.pszczolowska@gmail.com (M.P.);; 2Faculty of Health Sciences, Wroclaw Medical University, Ul. Kazimierza Bartla 5, 51-618 Wrocław, Poland; 3Clinic of Psychiatry, Department of Psychiatry, Wroclaw Medical University, Ludwika Pasteura 10, 50-367 Wrocław, Poland

**Keywords:** chronic traumatic encephalopathy, traumatic brain injury, Alzheimer’s Disease, oxidative stress, dementia, cognitive skills

## Abstract

This editorial investigates chronic traumatic encephalopathy (CTE) as a course of Alzheimer’s disease (AD). CTE is a debilitating neurodegenerative disease that is the result of repeated mild traumatic brain injury (TBI). Many epidemiological studies show that experiencing a TBI in early or middle life is associated with an increased risk of dementia later in life. Chronic traumatic encephalopathy (CTE) and Alzheimer’s disease (AD) present a series of similar neuropathological features that were investigated in this work like recombinant tau into filaments or the accumulation and aggregation of Aβ protein. However, these two conditions differ from each other in brain–blood barrier damage. The purpose of this review was to evaluate information about CTE and AD from various articles, focusing especially on new therapeutic possibilities for the improvement in cognitive skills.

## 1. Methodology Based on Preferred Presenting Items for Systematic Reviews and Meta-Analyses (PRISMA)

In this research study, the electronic database searches were focused on chronic traumatic encephalopathy and traumatic brain injury, as well as their connections and common points with Alzheimer’s Disease. This article is based on original articles, meta-analyses, characteristic cases, and systematic reviews that describe the above issues. Starting with studies from PubMed, the searched keywords were “chronic traumatic encephalopathy”, “traumatic brain injury”, “Alzheimer’s Disease”, “dementia”, and “cognitive skills”, mostly in a period from 2010 to February 2024. Additionally, articles related to the topics discussed and important publications from the past were included. As for the excluded articles, those that did not directly refer to the topic discussed, were outdated or too limited, and did not typically fit the context of the work were rejected.

## 2. Introduction

Chronic traumatic encephalopathy (CTE) is a debilitating neurodegenerative disease that results from repeated mild traumatic brain injury (TBI). It is increasingly seen in athletes [1]. Many epidemiological studies show that experiencing a TBI in early or middle life is associated with an increased risk of dementia later in life. Multiple mild TBIs in professional boxers have been observed to be associated with a high risk of chronic traumatic encephalopathy (CTE), a type of dementia with distinctive clinical and pathological features [2]. Chronic traumatic encephalopathy (CTE) and Alzheimer’s disease (AD) share several similar neuropathological features. These are primarily neurofibrillary tangles and hyperphosphorylated tau, but these entities have been described separately. New research shows that CTE and AD can occur in the same patients [3]. The neurofibrillary tangles found in the cortex in AD and CTE have different distributions. In AD, tangles in layers V and VI are visible in the deeper cortex with a laminar distribution.

In contrast, tangles may be perivascular in CTE and occur mainly in the more superficial layers II and III. Neurofibrillary tangles occur in the substantia nigra of people with CTE but are rare in AD. In AD, confusion appears most frequently in CA1; in CTE, confusion is seen in the horn of Ammon (CA1-CA4) [3]. 

Cases of CTE have been reported to coexist with Alzheimer’s disease (AD) pathology. In the cohort of CTE patients described by Stein and colleagues, amyloid β peptide (Aβ) deposition in the form of diffuse or neuritic plaque was found in 52% of cases [4]. The diagnosis of CTE is made post-mortem, but specific medical tools can identify characteristic features of CTE based on serum, clinical history, cerebrospinal fluid, and neuroimaging biomarkers. Knowledge of tau isoform processing mechanisms may lead to new treatments targeting proteases. Their goal is to prevent the formation of toxic tau fragments in neurodegenerative diseases of tauopathy [5]. CTE is compared to other known neurodegenerative entities, including Alzheimer’s disease and dementia with Lewy bodies [5]. Understanding the physiological processes in CTE and AD will allow us to compare these diseases and learn about their possible impact on each other.

## 3. Alzheimer’s Disease—Basic Facts

Alzheimer’s Disease (AD) belongs to the group of neurodegenerative diseases of the central nervous system. It is the most common cause of dementia (60-80% of cases), and it is expected that the number of patients with this diagnosis will increase due to the growing proportion of people who live beyond the age of sixty-five [6,7]. In the United States, Alzheimer’s remains the fifth-leading cause of death among people aged 65 and older [6].

There are many risk factors for AD and, among others, they are age, genetics (APOE-e4, Down syndrome, genetic mutations of the APP gene and genes for presenilin-1 and presenilin-2, and a first-degree relative with AD), smoking, obesity, hypertension, high cholesterol levels, diabetes, diet, low physical activity, lower socioeconomic status, and poor sleep quality. There have also been protective factors found, such as more years of formal education and remaining socially and mentally active throughout life, a heart-healthy diet, for example Mediterranean, Dietary Approaches to Stop Hypertension (DASH), or Mediterranean–DASH Intervention for Neurodegenerative Delay (MIND), and physical activity [6].

AD pathogenesis is based on the accumulation of the amyloid-beta (Aβ) outside of the cells and hyperphosphorylated tau proteins inside them. Those processes interfere with transmission between synapses and block the transportation of nutrients and molecules nourishing neurons. Another essential pathology is the degeneration of the cholinergic neurons in the nucleus basalis of Meynert. Amyloid deposits activate proinflammatory cytokines, which lead to chronic inflammation and exacerbation of the neurodegeneration process [8]. Many studies have found that vascular pathology and age-related changes in the blood vessels, such as brain–blood barrier (BBB) dysfunction or hypoperfusion and hypoxia, may play their part in AD’s pathogenesis [9,10].

Primary symptoms of AD are memory, language, and thinking problems. Personality, mood, and behavior changes, such as wandering, are also observed. With time, the impairment progresses, affecting other functions, including walking or swallowing. Patients with AD live on an average for four to eight years after diagnosis [6].

The diagnosis of AD is made based on DSM-5 criteria, which include meeting the requirements of major or mild neurocognitive disorder, insidious onset, and gradual progression of at least one cognitive disorder. Probable AD is diagnosed if there is evidence of a causative AD genetic mutation or if all of the following are present: evidence of a decline in memory and learning and at least one other cognitive domain, gradual decline in cognition, and no evidence of mixed etiology [11]. ICD-11 criteria are also used in the diagnosis. They include meeting all diagnostic requirements for dementia, the presumption of AD based on clinical assessment, standardized cognitive and neuropsychological testing, neuroimaging, genetic and medical tests, family history, and gradual onset with progressing memory and word-finding difficulties and mild functional impairment [12]. Additional tests can be used, such as TK, MRI, FDG amyloid PET, or CSF examination. TK and MRI can show hippocampal and cortical atrophy in temporal and parietal regions, whereas the changes in CSF include low Aβ and high tau and phospho-tau levels [7,13]. An essential part of the diagnostic process is also excluding other probable causes of dementia, such as vitamin B12 deficiency, hypothyroidism, HIV, or depression [13]. Current strategies in the pharmacological treatment of AD include acetylcholinesterase inhibitors, donepezil, rivastigmine, and galantamine, which improve cognition, and an NMDA receptor antagonist—memantine, which is used in moderate and severe forms of AD [8]. New therapies involve monoclonal antibodies such as aducanumab and gantenerumab, which target Aβ [14].

## 4. Chronic Traumatic Encephalopathy

Chronic traumatic encephalopathy (CTE) is a progressive, neurodegenerative condition that can be revealed after prolonged exposure to repeated episodes of blunt head trauma. CTE manifests itself as persistent cognitive and neuropsychiatric symptoms [15,16,17].

The discourse of CTE was first started in 1928 by Harrison Stanford Martland using the term “punch-drunk”. His observation was related to the characteristic neuropsychiatric syndrome, which some boxers presented after being exposed to repetitive traumatic brain injuries (TBI) [17,18,19]. From that point, more and more cases and studies were described. These studies featured scrupulously detailed variations, such as cortical atrophy, ventricular enlargement, fenestrated septum pellucidum, flattening of parts of the nervous system (e.g., the corpus callosum), substantia nigra depigmentation, cerebellar fibrosis, neuronal loss, and gliosis. There were also findings in neurons of the medial temporal lobe, cortex, and brainstem of argyrophilic neurofibrillary tangles (NFTs) [20]. As a result of all of these explorations and analyses, the neuropathological criteria for CTE were defined by the NINDS/NIBIB in 2016, with diagnostic refinements made in 2021 [21].

In Martland’s works, “punch-drunk” referred to specific symptoms, mood issues, and forms of behavior (described, e.g., as “goofy” or “slug nutty”) [21,22]. However, nowadays, CTE refers precisely to the neuropathological diagnosis in tissues and can be definitively diagnosed only by post-mortem neuropathological examination [21].

Starting from epidemiology, CTE is mainly associated with professional and amateur athletes, especially American football players and boxers, but it should not only be. Plenty of different groups of people are exposed to repetitive head impacts (RHIs), also known as repetitive traumatic brain injuries (TBIs), defined as cumulative exposure to recurrent concussive and subconcussive events. These injuries can be caused through contact and collision sports at any age, military service, severe epilepsy attacks, or being the victim of domestic violence [16,17,23,24].

### Molecular Mechanisms of Tau Protein in CTE and AD

From a molecular point of view, CTE is a tauopathy with progressive accumulation of the tau protein observed even in middle-aged patients (the first significant findings were made in American football players in 2005) [19,23,25]. The characteristic pathognomonic lesions of a perivascular accumulation of neuronal phosphorylated tau (p-Tau) are described. These neuropathological findings can initially localize in mechanical trauma regions and then spread into the whole structure through an inflammatory response. The pathology of this process is exceptional, as these lesions are unlike any observed tauopathy [16,21,23,24] and will be precisely described in the subsequent paragraph.

Referring to TBIs mentioned at the beginning, it is worth highlighting that a traumatic brain injury (TBI) increases the risk of developing neurodegenerative diseases (NDs) and neurocognitive disorders. A TBI triggers extensive epigenetic modifications somehow. However, the precise molecular pathways still need to be fully elucidated and require more research. While taking this approach, other NDs (such as Alzheimer’s or Parkinson’s Diseases (AD and PD)) are worth mentioning, as they are strongly connected with the epigenetic changes of the comprehensively described mechanisms [23,26]. The present data suggest that a TBI is a potent risk factor for these NDs (CTE, AD, and PD) due to, e.g., persistent neuroinflammation, but most theories are hypothetical theses [23]. In conclusion, TBIs are being considered a cause of significant preventable neurodegeneration [16].

Lastly, there is a controversial side to CTE. As featured in Randolph’s publication in 2018, there are also individual opinions that CTE is not an actual disease. Some scientists point out the lack of specific, verifiable data and accept it on the case study level with doubtful pathomechanisms. Additionally, the all-cause mortality rate as well as the suicide rate are about less than half of those expected in the presumptions of CTE [16,27]. To summarize, this area requires lots of confirmed and solid data. Nevertheless, there is a connection between TBIs and CTE and NDs [16,23].

## 5. Assembly of Recombinant Tau into Filaments (CTE and AD)

AD is a neurodegenerative disease, with intracellular neurofibrillary tangles of hyperphosphorylated tau and extracellular amyloid plaques composed of amyloid-β (Aβ) being characteristic in this condition [28]. Most age-related neurodegenerative diseases involve assembling small numbers of soluble proteins into insoluble amyloid fibrils. Of these proteins, tau is the most commonly affected. Tau is made into fibers in many diseases, including Alzheimer’s disease (AD), chronic traumatic encephalopathy (CTE), globular glial tauopathy (GGT), corticobasal degeneration (CBD), or Pick’s disease (PID) [29]. 

### 5.1. Tau Protein Structure and Isoforms

Tau is expressed mainly in the central and peripheral nervous system and is most abundant in the axons of nerve cells. It is divided into an N-terminal projection domain, a proline-rich region, a repetitive region, and a C-terminal domain [30]. Research suggests that tau in the form of oligomers, or more specifically in the form of filaments, is considered a neurotoxin. Before medical symptoms appear, tau oligomers are sequestered in the early stages of the disease. Morphologically, tau oligomers take the form of a sphere using atomic force microscopy. They fuse with two or more tau molecules between 6 and 20 nm in size [31]. 

In the human brain, six different isoforms are generated by alternative mRNA splicing mechanisms, which range from 45 to 65 kDa in length according to SDS gels. The isoforms differ in the presence or absence of the R2 domain, one of four partially repeating regions of the microtubule-binding domain (MTBR) designated R1, R2, R3, and R4, and the presence or absence of two 29 aa amino-terminal inserts encoded by exons 2 and 3 that are referred to as 0N, 1N, or 2N tau [32]. 

### 5.2. Pathological Tau Phosphorylation

In CTE, the deposition of hyperphosphorylated tau usually begins in the perivascular furrow area in the cortex and then spreads unevenly in the cortex. In contrast, AD shows a diffuse distribution of hyperphosphorylated tau in cortical regions [33]. Three tau phosphorylation sites, Thr231, Thr181, and Ser199, serve as biomarkers of AD. Hyperphosphorylation of tau at Thr212/Thr231/Ser262 or Ser199/Ser202/Thr205 can cause cell death because it leads to microtubule instability [34]. Phosphorylation at Thr175, Tor 231, Ser199, and Ser422 has been reported in CTE [35]. Phosphorylation of tau at Thr231 induces a cis–trans conformational change in p-tau. Within hours after a TBI, the cis p-tau conformation appears in neurons before tau oligomers, prefibrillar tangles, and NFTs are formed and cause damage to axons [36]. We can infer that cis p-tau has a function that drives neurodegeneration. 

The assembly of tau into filaments can be initiated or accelerated by adding granules. Human tauopathies may spread in the brain. It has been studied that tau assemblies, when applied extracellularly, can cause aggregates to form, and then tau aggregates are laid [30]. The spread of the tau protein requires seeding and the capture and release of aggregates. Expressed tau can only be buried if it is capable of aggregation. Aggregation inhibitors may be responsible for reducing tau-induced seeding and spreading. Tau induces them. Many studies believe that monomeric tau cannot form aggregates [30]. 

Individual diseases differ in tau folds, but tau filaments from individuals with a given disease have the same structures. Identical protofilaments can arrange themselves differently to form distinct filaments, so-called ultrastructural polymorphs. In AD, two protofilaments join symmetrically to form paired helical filaments (PHFs) or asymmetrically to form straight filaments (SFs). Alzheimer’s and CTE folds consist virtually entirely of R3 and R4. The distribution of protofilament fold cores explains the composition of tau filament isoforms [37]. Recombinant full-length tau is highly soluble [38]. Tau in Alzheimer’s Disease also differs from normal tau by its very high degree of phosphorylation. The high degree of phosphorylation predisposes tau to aggregation in PHFs, but no conclusive confirmatory studies exist. Phosphorylation, particularly of the repeat domain, clearly regulates tau–microtubule interactions [39]. Moreover, while tau filaments exhibit diverse configurations across individual diseases, the involvement of Aβ protein emerges as a crucial factor in the pathogenesis of Alzheimer’s disease (AD) and chronic traumatic encephalopathy (CTE). Understanding the structural intricacies of tau filaments sheds light on the interplay between tau pathology and Aβ accumulation, elucidating the complex mechanisms underlying neurodegeneration.

## 6. Aβ pathology in AD and CTE

The Aβ protein plays a crucial part in AD pathogenesis. Its accumulation and aggregation outside the cells trigger AD progression [8]. It consists of 39–40 amino acids, and its biogenesis involves proteolysis of the APP protein, i.e., a transmembrane protein. There are two paths of APP degradation: amyloidogenic and nonamyloidogenic. In the nonamyloidogenic pathway, secretases of alpha and gamma are engaged, and they lead to the formation of soluble products. In the pathogenesis of AD, APP is degraded through the cleavage of beta- and gamma-secretase, leading to the generation of insoluble Aβ, which aggregates into plaques [40], as illustrated in Figure 1.

Aβ oligomers have many neurotoxic effects. Aβ deposits activate various Toll-like receptors (TLRs) and coreceptors and cytokines of IL-1β family production, leading to chronic inflammation and consequently to an impairment in dendritic spines and disruption in the microglial clearance of Aβ. The synthesis of nitric oxide (NO) and the activation of CDKs is also enhanced, which further increases the peptide’s ability to aggregate and suppress synaptic plasticity, as well as increasing the hyperphosphorylation of tau [8]. The aggregation of Aβ also promotes free radicals such as reactive oxidative species (ROS) through binding with metal ions, such as zinc and copper [41]. Oxidative changes in Aβ caused by ROS make its clearance even less effective. Aβ-induced oxidative stress also leads to the oxidation of proteins and lipids, causing cell membrane defects of DNA, resulting in its damage [40]. All of the effects are presented in Figure 2.

Aβ also aggregates in cerebral and leptomeningeal blood vessels, leading to cerebral amyloid angiopathy (CAA). The result of this pathology is the destruction of blood vessel walls, which increases the risk of either microhemorrhages or extensive lobar hemorrhages [42,43]. Another consequence of CAA can be leukoencephalopathy and CAA-related inflammation or angiitis. The described pathologies can cause a stroke and lead to further progression of dementia and cognitive impairment [43].

When it comes to chronic traumatic encephalopathy (CTE), Aβ can be found in the brains of about half of diagnosed patients, but the role of its accumulation in CTE is still not well understood [44]. It appears that in CTE patients, age-dependent Aβ deposition is accelerated compared to the general population, and they develop Aβ accumulation at a younger age [45,46]. Furthermore, in CTE, damage to cerebral blood vessels is observed, and it is associated with CAA [7]. Aβ presence is also connected to CTE progression. Studies show that Aβ is not present in stage I of CTE; in stage II, it is already observed in 19% of patients older than 50, and the number grows significantly in stages III and IV [8]. It was also observed that it increased the stage of CTE tauopathy. It also accelerated the development of dementia. The amyloid plaque localization is also different from that in AD—it is more peripheral, suggesting that its accumulation’s pathogenesis is based mainly on impaired CSF clearance of Aβ [7]. It was also found that the presence of APOE ε4, one of the significant risk factors for AD, leads to greater accumulation of Aβ after a traumatic brain injury (TBI) and further increases the likelihood of its deposition in CTE [6].

The role of Aβ in CTE is still not fully known. Therefore, studies on this topic are needed to find more possible connections between CTE and AD in this field, which could lead to a better understanding of both diseases.

## 7. Endothelial Cell and BBB Damage in CTE and AD

The BBB is a physical barrier consisting of endothelial cells (ECs), which are connected tightly and communicate with themselves through specific junctional proteins. These cells act as the proper guards of the barrier; they maintain the integrity and regulate the movement of plasma proteins, ions, and all nutrients and toxins in the system. ECs can manage the expression of various transporters capable of different types of molecular transport (e.g., solute carrier-mediated, receptor-mediated, active efflux, and ion) [47,48,49]. The construction of the BBB is presented in Figure 3.

Regarding BBB construction, ECs are attached to a basement membrane of specific collagen (type IV), fibronectin, and laminin. Not only are ECs surrounded by a basement membrane, but pericytes also are. Pericytes can express specialized particles or partitions in signaling pathways as well, and thanks to their perivascular location, pericytes are an essential part of the immune response of the nervous system [47,48,49].

Lastly, astrocytes, the characteristic star-shaped glial cells, must be mentioned. By connecting with their endfeet wrapping around the capillary walls, astrocytes connect and communicate with the brain’s capillaries and neurons, constituting a linkage between the blood and the brain [47,48,49].

All of these things considered, the small arterial cross-sections are surrounded by these three different cell types, which determine the pivotal role of the BBB [48,50]. From this point, the image of the solid structure that surrounds the minor arterials is made. However, besides the isolation and the protection of the brain from different agents such as toxins or microorganisms, the BBB plays a vital role in the communication between these two worlds. EDc are strictly connected by specialized tight junctions, and even though the substance transport is minimal, it is worth emphasizing that oxygen and carbon dioxide can diffuse across the barrier without hindrance [48,49]. Additionally, special transport is performed for a whole variety of required agents, e.g., transmembrane diffusion for small lipid-soluble molecules, low-weight compounds (<400 Da), or ethanol (due to its low-hydrogen bond structure); solute carrier-mediated transport (CMT) for compounds such as carbohydrates, amino acids, fatty acids, monocarboxylic acids, nucleotides, hormones, vitamins, and organic ions; receptor-mediated transcytosis (RMT) for peptides, polypeptides, or even proteins; and the sodium pump (Na+ and K+-ATPase) for sodium and potassium flux [48,51,52].

Knowing the BBB’s major components and functions, it is perfectly understandable that any BBB dysfunction can disrupt the complexity of brain homeostasis and reduce its protective effect on the central nervous system (CNS). Initially, these changes were attributed to vascular dementia. However, a new approach has appeared with the growing amount of data [53]. These changes can be detrimental to a variety of NDs. The ongoing endothelial pathology can contribute to the damage of the nervous structure and to Aβ and tau pathology, as described above [48,50,53].

This paragraph briefly specifies the BBB damage in NDs, such as AD and CTE. In the following parts, mitochondrial dysfunction in the vasculature and cerebrovascular inflammation will be discussed.

We start with the most apparent and visible way to observe BBB damage—the section. Described analyses of post-mortem tissues from patients with AD and other NDs demonstrate brain capillary leakages. Additionally, the accumulation of white or red blood cells from invalid infiltration and the degeneration of pericytes and EDc are observed with various molecular changes in the BBB and all CNS structures [48]. Moreover, when comparing AD with CTE, the results are adequate to confirm their dissimilar pathophysiology. In AD, the most prominent changes are the accumulation of fibrinogen, thrombin, albumin, IgG, and hemosiderin in the cortex and hippocampus. Mentioned proteins are often found to be characteristic of Aβ. In CTE, the perivascular macrophages and histiocytes with hemosiderin can be noticed [48,54,55].

While focusing on the post-mortem studies mentioned and characterized earlier, Aβ is a noticeable factor of BBB damage. The accumulation of Aβ in the cerebral meninges and smaller cerebral vessels, called cerebral amyloid angiopathy (CAA), is featured as the major BBB disruptor and a significant hallmark of AD. Moreover, even on the AD prediction level, vascular biomarkers can be detected before the noticeable CAA and Aβ deposits, leading to a cause-and-effect sequence conclusion [56].

Successively, introduction to the two-hit vascular hypothesis of AD is required. In this approach, the initial step of the pathological pathway, following genetic predisposition, is specifically cerebral blood vessel damage. From this point, BBB dysfunction starts and leads to all of the neurodegenerative processes described in the previous paragraphs. In addition to the first step, lifestyle and environment are mentioned, where variable factors can also influence Aβ propagation, which determines the second step and the development of AD [51,57]. This hypothesis is illustrated in Figure 4.

Returning to other NDs, CTE mainly, BBB disruptions are also found. In the injury-exposed regions of the brain, a high density of perivascular p-Tau is described, with immunoreactivity signs of interruptions in the tight junctions (especially its components such as claudin-5 and zonula occludens-1) and with enhanced extravasation of endogenous blood components such as fibrinogen and IgG [58,59].

Damage of the BBB is a complex issue. It is worth remembering that these neurodegenerative changes limit or demolish the physiological processes and impact the pathophysiological ones. The damage also results in toxins and all kinds of xenobiotic accumulation. These xenobiotics, such as food additives, allergens, pollution, and drugs, also react with neurogenic brain functions and structures, contributing to disease development [48]. Moreover, the disturbed mechanism of cleansing the cerebral vascular space leads to accelerated Aβ accumulation connected with AD. From this point, another chain of events is made, where the reduced blood flow and increased Aβ levels are described due to the BBB damage. These changes successively promote pathological p-Tau pathways [60].

Approaching the end of this paragraph, it is important to mention that another step of xenobiotic accumulation is a valid aspect of neurodegeneration. Apart from p-Tau formation, an increasing inflammatory response due to the activation of astrocytes and microglia is also observed. The stormy secretion of cytokines and chemokines occurs in vascular and cerebral tissues, contributing to progressive neurodegenerative changes. This cerebrovascular inflammation will be discussed later [48].

## 8. Mitochondrial Dysfunction in the Vasculature in CTE and AD

Transitioning from the discussion of xenobiotic accumulation and inflammatory responses in neurodegeneration, we now turn our focus to another critical aspect: mitochondrial dysfunction in the vasculature in both CTE and AD. Traumatic brain injury can instigate chronic traumatic encephalopathy by disrupting blood flow within the nervous system, leading to ischemia and subsequent elevation of reactive oxygen species (ROS) levels. These molecules are produced in small amounts in healthy mitochondria; however, due to stress conditions, their level is increased. This may lead to many threatening actions in the cell, such as the induction of apoptosis through lipid peroxidation. Interestingly, ROS may release cytochrome C (CytC), which also may start apoptosis [61] or activate Nuclear factor-kappaB (NF-κB), which regulates the immune response. However, increased levels of NF-κB were discovered in neurons that were dying [61]. In a study on rodents, cerebral ischemia resulted in the activation of the NF-κB/MAPK/JNK pathway [62]. Moreover, elevated ROS levels may lead to DNA destruction like base damage and single-strand breaks (SSBs) [63] and mitochondrial permeabilization. 

While numerous studies prove that ischemic brain damage upregulates the protection against antioxidants [64], some research indicates the downregulation of this defense. Kaur et al.’s study described reduced antioxidant defense in the early hours of post-stroke [6]. Antioxidant protection mechanisms are considered inefficient in the defense against elevated ROS levels, leading to cell death [65].

What is more, mitochondrial dysfunction due to brain injury is associated with an increased level of nitric oxide (NO). NO may regulate the functioning of this organelle through S-nitrosylation of mitochondrial proteins, like Receptor-interacting protein 3 (RIP3), a molecule sensor regulating cell apoptosis and necrosis [64]. This molecule can modulate apoptosis at several levels, such as blocking the apoptosome [66], inhibiting caspases [67], or influencing other cellular targets [68]. It was also proven that NO can cause mitochondria to commence cell death [69].

The ratio of NAD/NADH is critical when regulating many processes in the cell. As a result of severe trauma, poly (ADP-ribose) polymerase-1 (PARP-1) is the most active enzyme because it can promote DNA repair [66]. Activated PARP-1 leads to a significant decrease in intracellular NAD, especially under conditions of metabolic stress, while the production of NAD is limited due to the reduction in intracellular ATP [67]. 

Mitochondrial fragmentation and dendritic damage may be observed as a response to severe injury. Mitochondria may also change their structure—from a thin and long morphology to being swollen and fragmented. Ischemic stroke or brain trauma may affect mitochondria via two different mechanisms—they can protect neurons by adsorbing excessive Ca2+ and increasing levels of ATP or they can discharge proapoptotic factors that lead to cell death [70]. Proteins that mediate mitochondrial fission and fusion, such as Drp1, Opa1, and Mfn1/, are necessary for ischemic neuronal death [71,72]. 

Additionally, previous studies demonstrate that amyloid-β plaques enhance mitochondrial fission by increasing the generation of NO, which promotes S-nitrosylation of Drp1 [73,74]. These findings suggest that disturbances in mitochondrial activity are similar to traumatic brain injuries and AD. Furthermore, mitochondria play a crucial role in various astrocyte processes, emphasizing the importance of balanced mitochondrial dynamics in maintaining brain health [75,76].

## 9. Cerebrovascular Inflammation in CTE and AD

Building upon the link between traumatic brain injury (TBI) and Alzheimer’s disease (AD), recent research by Tajiri et al. [77] confirms a pathological connection between the two conditions. In their experimental model, TBI significantly impacted the cognitive performance of AD-transgenic mice and nontransgenic (NT) mice trained in spatial memory tasks. Two and six weeks post-TBI, AD mice exhibited worsened performance compared to AD mice without a TBI and NT mice, regardless of TBI exposure [78].

The aftermath of a TBI involves damage to various brain structures, including neurons, axons, glia, and blood vessels, initiating numerous biochemical pathways that lead to cell death and neurodegeneration [79]. This primary injury not only affects the surrounding tissues directly but also triggers a cascade of neurodegeneration that can spread throughout the brain. The immediate inflammatory response, characterized by microglial activation, persists for at least one year post-injury, as evidenced by studies in rats and rhesus monkeys [80,81]. These findings suggest that the onset of a progressive degenerative process is associated with TBIs, potentially contributing to the later development of AD [81]. Microglial response to the injury is immediate [82], lasting for at least 1 year, which is proven by a study on rats [77]. There was significant tissue loss and ventriculomegaly in the hemisphere ipsilateral to the injured brain. The results of this study suggest the commencement of a progressive degenerative process connected to the occurrence of a TBI. Similar conclusions are drawn in a study on the rhesus monkey, whose microglia and macrophages were still activated one year after TBI [80]. 

A secondary cell death mechanism resulting from TBI is the secretion by injured cells of a neurotransmitter called intracellular glutamate. It is secreted into extracellular spaces [83], leading to cell necrosis and apoptosis by activating protein phosphatases, phospholipases, and endonucleases to fragment DNA [84]. 

Secondary cell death following a TBI may also be caused by neuroinflammation [85]. Some research claims that this process may last for many years and predispose patients to the development of AD [82]. Immune cells, microglia, cytokines, chemokines, and other inflammatory molecules are responsible for immunological responses [86]. A recent study has revealed increased levels of cytokine CCL11 in the frontal cortex in the brains of CTE patients as opposed to patients with AD [87]. Excessive activation leads to the upregulation of proinflammatory factors and increases the amount of neurotoxic molecules and free radicals, activating additional cell death mechanisms [78]. What is more, excessive expression of major histocompatibility complex class II (MHCII+) also leads to neurodegeneration [88]. 

Intriguingly, microglia exhibit two distinct phenotypes: M1, which increases the production of proinflammatory cytokines while decreasing anti-inflammatory cytokines, and M2, which stimulates inflammation by reducing proinflammatory cytokines and promoting anti-inflammatory molecules [89].

Similarly, astrocytes play a dual role in inflammation, capable of both increasing and decreasing inflammatory responses. Surface glycoproteins like the scavenger receptor activate pathways involved in neuroinflammation and the development of neurodegenerative diseases [90]. 

## 10. Treatment Methods of CTE

As primary injury can only be prevented by avoiding traumatic brain injuries (TBIs), the therapeutic focus for chronic traumatic encephalopathy (CTE) shifts toward mitigating events occurring during secondary cell death. The primary goal is to delay the onset of neuroinflammation [91]. One promising avenue in this regard is minocycline, a tetracycline derivative known to pass through the blood–brain barrier. Research indicates that minocycline reduces the concentration of proinflammatory cytokines and chemokines by modulating inflammatory regulators. Animal studies have shown normalization of serum and tissue levels of various inflammatory, vascular, neuronal, and glial markers, with animals treated with minocycline demonstrating improved neurological outcomes [92,93]. Nonetheless, further investigations are warranted to ascertain its suitability for humans with TBIs.

Another promising drug is melatonin, a hormone from the pineal gland. This hormone may inhibit microglial activation and reduce proinflammatory cytokine concentrations, limiting microglial activation. A study on mice revealed that the nuclear factor erythroid 2-related factor 2 and antioxidant-responsive element (Nrf2-ARE), an antioxidant, is transferred from the cytoplasm to the nucleus. It activates the Nrf2-ARE signaling pathway, which protects the cell from many pathological processes, including oxidative stress [94].

Another potential treatment option is statins, which have limited microglial and astrocyte activation. They reduce proinflammatory cytokines IL-1 β and TNF-α and intracellular adhesion molecules. Drugs like rosuvastatin limit brain superoxide production and inhibit factor-kappa B (NF-κB) activation. This substance has also promoted the activation of microglial cells after SAH [95]. Preclinical studies have proven its effectiveness in animals and humans in treating TBIs [96].

Stem cells are a hope for patients with TBIs due to their high degree of proliferation. This is the reason why the administration of stem cells improves cognitive skills, limits inflammation, and decreases cell death [97,98]. A special type called mesenchymal stem cells (MSCs) is used especially for TBI treatment because it strengthens the BBB. It does so by limiting the production of several chemokines, which results in decreasing inflammation. Decreasing the level of CXCL2 occurs by limiting neutrophil infiltration. Limiting CCL2 stops immune cell relocation, and the blocking of RANTES stops T-cell activation [99,100].

Interestingly, stem cells may be administrated not only to replace the damaged tissue but also to modulate the inflammatory response [101]. Stem cells may release a paracrine set of regulating molecules like cytokines, chemokines, and growth factors [102]. Those molecules may be released through exocytosis or by transport proteins and extracellular vesicles [103]. This is a new treatment possibility for neurodegenerative diseases, for example AD. There was a study conducted in which MSCs were intravenously injected into a transgenic mouse model of AD. After six weeks of injections, mice improved spatial learning, and there was a decrease in Aβ plaques in the brain [104]. MSCs may also be activated to express TSG-6, a multifunctional protein that modulates inflammation. This molecule built of a hyaluronan-binding Link domain and CUB module may inhibit neutrophil migration, enhancing the inhibiting activity of some enzymes that disturb the inflammation process [105,106,107]. 

The administration of granulocyte colony-stimulating factor (G-CSF) along with stem cells improves the treatment outcome due to the improvement in the survival rate of the stem cells. G-CSF can cross the BBB. This combined therapy increases angiogenesis in the brain and decreases inflammation and apoptosis [81].

Due to the mitochondrial imbalance which appears as a result of a TBI, some antioxidants have been proposed as a treatment option. Molecules like CoQ10 or MitoQ presented positive trial results [108,109]. Other therapeutic options are nicotinamide riboside (NR) and nicotinamide mononucleotide (NMN), which are NAD+ precursors [110]. 

One of the new ideas for treating AD is to use inhibiting substances like capsaicin, coenzyme Q2, mucidin, and stigmatellin, the largest multi-subunit complex that contributes to creating reactive oxygen species (ROS) [111].

Pharmacological therapy may be divided into the areas of cognition that it aims to improve. For memory impairment, the most suitable therapy is cholinesterase inhibitors [112,113]. Apathy, which is often a symptom for patients with dementia, may be treated with levodopa formulations, dopamine agonists, and stimulants [114]. Stimulants may be also useful for impaired attention treatment, which is common in CTE patients. The most commonly used drug is methylphenidate. This drug was found to decrease mental fatigue and improve mental processing speed, attention, and working memory [115]. Depression and anxiety are common in CTE patients. Using antidepressants in treating those conditions may also be beneficial in improving cognitive skills [116,117]. Agitation, which also occurs in CTE patients, is treated with different drugs depending on the cause of this pathology. Anxiety caused by agitation may be treated with sertraline or escitalopram. Night-time agitation, which is caused by sleep disturbances, is primarily treated with sleep hygiene, followed by low doses of risperidone [118].

It is very important to note that nonpharmacologic or combination therapy may be the first-line treatment for CTE. One of the disease treatments is also exercising. Improving blood circulation in the brain thanks to aerobic exercise has been described in some studies [19,20]. The positive influence of aerobic exercise on neuroplasticity and the subsequent increase in neurogenesis was also investigated [21].

Another option for patients suffering from CTE is a Mediterranean diet. This type of diet is said to be balanced in the best possible way, providing the patient with the highest amount of nutrients and, at the same time, the lowest dose of saturated fats [22].

Cognitive rehabilitation is said to improve daily life quality, but not to treat the disease. This kind of therapy is a set of interventions that help the patient’s cognitive skills improve. It is beneficial at any time after the injury, in both mild and severe injuries. It is very important that at the beginning of the treatment process, there is a precise examination of cognitive skills performed to obtain the most accurate treatment plan [23]. This form of treatment was proposed in the last century; however, new inventions in technology have allowed us to obtain better results [24].

As a support for the main therapy, there may also be mood/behavior therapy administered. This would be beneficial for patients presenting depression or anxiety symptoms. Cognitive behavioral therapy (CBT) helps to create balanced cognitive patterns to avoid the stress and anxiety associated with psychological problems. CBT aims to teach functional and helpful behaviors. It is also believed that practicing mindfulness is helpful for some patients [25].

Some sources claim that occupational–ocular therapy (OOT) may improve cognitive skills in patients suffering from CTE which limits blurred or double vision. The help from certified ocular therapists may be beneficial in limiting other visual problems and improving visual processing and perceptual skills. Also, vestibular therapy may improve cognitive skills for patients whose hearing is impaired from a TBI [19]. 

Due to physical therapy, patients with impaired motor systems influence improvements, even in neurological functioning. Transcranial magnetic stimulation is believed to cause many beneficial results, the same as hyperbaric oxygen therapy [19]. For patients who suffer from cognitive impairment associated with neuroendocrine dysfunction, a new possibility for improvement is endocrine therapy. The hormonal problems caused by a TBI may be gonadotropin deficiency, growth hormone deficiency, secondary adrenal insufficiency, diabetes insipidus, inappropriate antidiuretic hormone secretion (SIADH), or thyroid-stimulating hormone deficiency [21]. Sometimes, this kind of hormonal disturbance may be only slightly intense and health care providers may not recognize the problem because of the brain injury itself. It is crucial to treat these conditions with proper hormone substitution.

As we look ahead to the future of CTE treatment, the focus lies on two main directions: better prevention of traumatic brain injuries (TBIs) and more effective strategies for reducing neuroinflammation. The ideal treatment would involve controlling the cells and molecular processes contributing to secondary cell death, including immune cells, astrocytes, cytokines, and chemokines. A deeper understanding of CTE pathology will undoubtedly inform future treatment modalities.

## 11. Imagining of CTE

Transitioning from the prospects of treatment, a critical aspect of diagnosing CTE involves neuropathological examination. This diagnostic approach allows for the visualization of aggregations of the hyperphosphorylated tau protein, which can be observed in both neurons and glial cells [119,120]. Specifically, the identification of cis p-Tau, a distinct isomer believed to be generated following traumatic brain injury, holds diagnostic significance [121]. Additionally, according to the Preliminary National Institute of Neurological Disorders and Stroke criteria, the presence of disordered neurites around small vessels in an irregular pattern at the depths of the cortical sulci is required for the pathological diagnosis of CTE [122].

Mild cases may not present any macroscopic changes but advanced stages of CTE are characterized by decreased brain weight, extension of the ventricles, and atrophy of the neuronal tissue, especially in the frontal and temporal lobes. In some of the most severe cases, atrophy of the thalamus and hypothalamus may be observed. In most cases, changes like septal fenestrations or depigmentation of the substantia nigra may be observed [15].

Together with the severity of p-tau neurodegeneration, the degree of axonal disintegrity increases [123]. There may also be abnormally phosphorylated TDP-43 proteins observed. This protein may be found in the cerebral cortex, the medial temporal lobe, the diencephalon, the brainstem, and even the spinal cord [124].

Positron emission tomography (PET) allows us to see changes in the brain taking place during a TBI. Post-mortem studies have revealed microglial activation using the ligand 3HPK11195, which was present even many years after the traumatic injury. In the analysis of astrocyte activity, which also helps to characterize the range of the brain damage, there may also be fluorescence microscopes [76]. Among the PET ligands used in CTE neuroimaging may also be Fluorodeoxyglucose (FDG), which measures brain metabolism and glucose uptake level. Florbetapir (18F-AV-45) indicates fibrillar beta-amyloid deposits, while Flortaucipir (T807, 18F-AV1451) binds to the paired helical filament (PHF) tau. 18F-FMZ is a ligand that is helpful in GABA metabolism observation and 11C-DPA-713 facilitates neuroinflammation level observation [125]. 

Magnetic resonance spectroscopy (MRS) is also useful in CTE imaging. It reflects on metabolite concentration, which allows us to observe brain metabolism. High lactate levels indicate hypoxia and impairment of perfusion, while choline (Cho) increases points at diffuse axonal injury [125].

Magnetic resonance imaging (MRI) delivers detailed brain morphology information. It is beneficial in quantifying volumetric changes in the patient’s brain. Conventional MR cannot visualize the microscopic damage, but those pathologies may be detected with diffusion tensor imaging (DTI), a noninvasive MRI method. This helps discover changes in the brain’s white matter, relying on the water’s diffusion properties [126]. A functional MRI (fMRI), which reflects cerebral blood flow in the brain, helps diagnose and observe the disease course [127]. 

Differentiating CTE from other neurodegenerative diseases, especially AD, is very important to ensure the best treatment options are chosen. Crucial to distinguishing CTE from AD is histopathological examination. AD shows a common and highly consistent pattern of distribution of neurofibrillary tangles (NFTs); on the contrary, the distribution of NFTs is more sporadic in CTE [128]. More characteristic of CTE is the presence of NFTS at the depths of sulci and the perivascular clustering of NFTs [129]. Interestingly, Tau distribution in CTE may reflect head injury patterns since the depths of the sulci are areas of stress concentration due to forces during a brain injury. Studies on contact-sport athletes revealed that pathological findings in the superior and dorsal–lateral frontal lobes in their brains matched frequent high-intensity impacts on the crown of their helmets [130]. The Aβ pathology in CTE patients is more significant in the depths of the sulci, contrary to AD patients, which had similar levels of Aβ plaque deposition both in the sulci and gyri [44].

## 12. Conclusions

The transition from discussions on neurobiological mechanisms to therapeutic possibilities underlines the interconnected nature of AD and CTE, two neurodegenerative diseases that exhibit both similarities and differences. A contradiction of the thesis stated in the title may be the fact that the β-helix region is differently confirmed in CTE patients’ brains compared to those of people suffering from AD. This different conformation is responsible for creating a hydrophobic cavity in tau filaments. What is more, there are varied protofilament interfaces in both conditions [131]. In the work that used multiplex ELISA, there were specific clusters for each tauopathy identified. There were different proteins characteristic of CTE and AD. CCL21 was the most specific for CTE and FLT3L for AD [132]. These facts are, based on some researchers’ arguments, contrary to the common pathology of the diseases.

By delving into topics such as cerebral vasculitis, Aβ pathology, and mitochondrial dysfunction in both conditions, this study highlights the intricate relationship between them. Moreover, it elucidates how neuropathological features characteristic of both diseases can coexist within a single individual, emphasizing the importance of dual diagnosis.

Understanding this interplay between neurotrauma and neurodegeneration is crucial as it offers insights into the impact of injury and age on disease development. Furthermore, by presenting treatment methods for CTE and exploring better biomarkers for detecting damage, this study paves the way for improved diagnosis and management strategies.

Moving forward, continued scientific inquiry into the relationship between CTE and AD holds promise for advancing our understanding of these overlapping human diseases. By uncovering new insights, researchers may ultimately contribute to the development of more effective diagnostic tools and therapeutic interventions for individuals affected by these debilitating conditions.

## Figures and Tables

**Figure 1 ijms-25-04639-f001:**
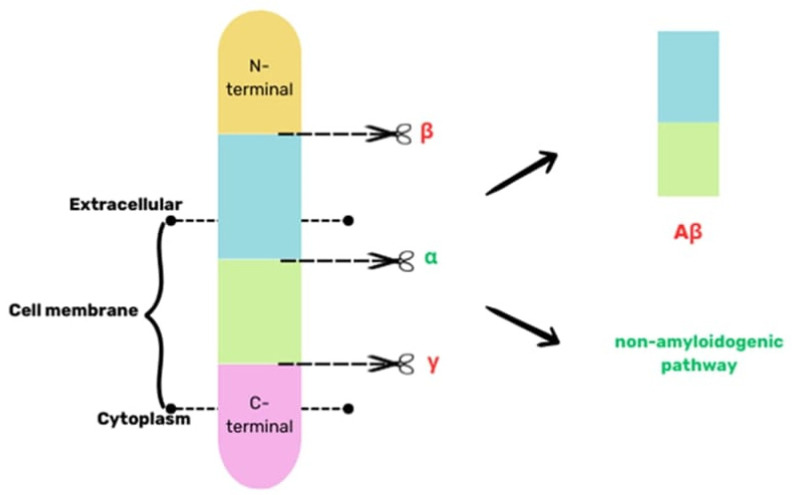
A simplified scheme of the biogenesis of Aβ based on [40].

**Figure 2 ijms-25-04639-f002:**
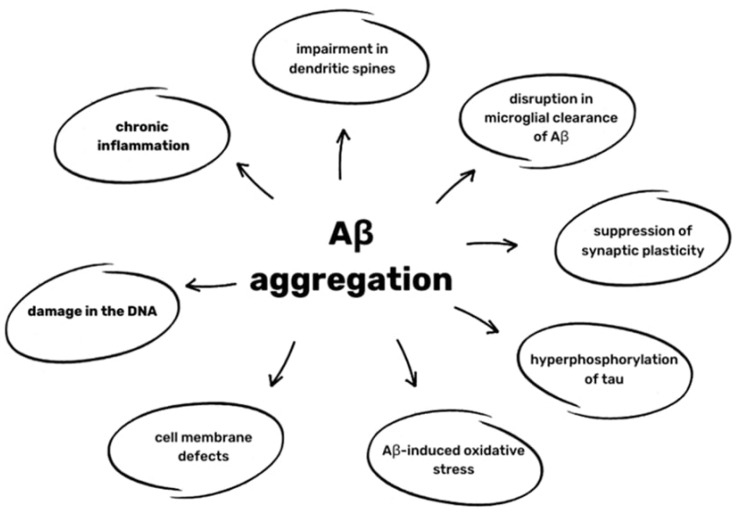
Effects of Aβ aggregation based on [8,40,41].

**Figure 3 ijms-25-04639-f003:**
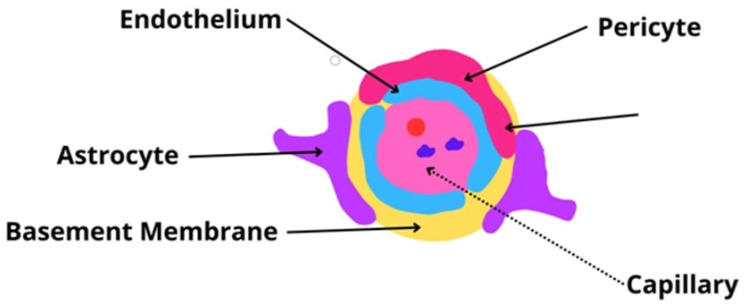
The blood–brain barrier—a representation image of the most crucial structure [1,2,3].

**Figure 4 ijms-25-04639-f004:**
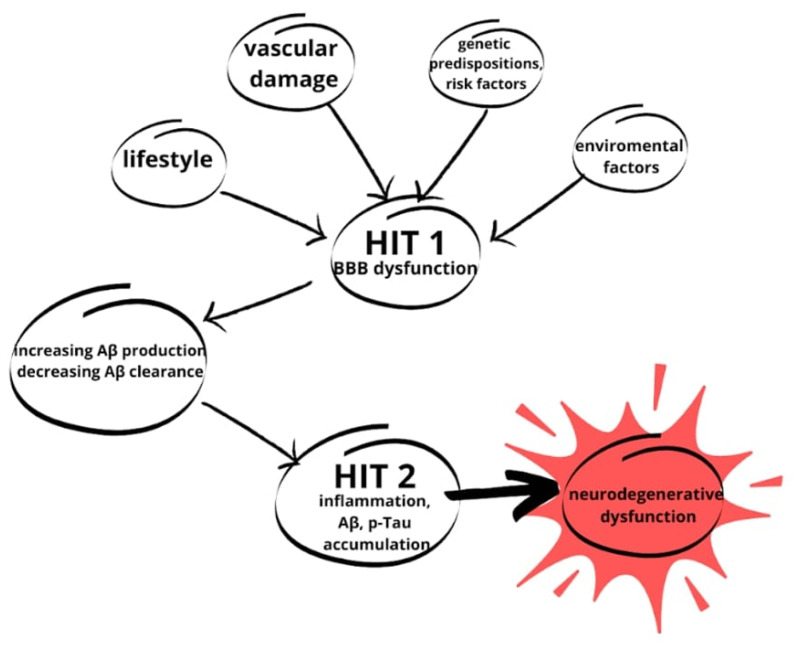
The two-hit hypothesis of AD. The first hit (HIT 1) is due to variables such as lifestyle, genetic predispositions and risk factors, environmental factors, and vascular damage described in this paper. The variables initiate a cascade of neurodegenerative changes. With BBB damage, the accumulation of Aβ starts, which strongly contributes to HIT 2. From HIT 2, through inflammation and Aβ and p-Tau accumulation, neurodegenerative changes lead to NDs (based on [51]).

## Data Availability

Not applicable.

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
