# Peer review of "Chronic Traumatic Encephalopathy as the Course of Alzheimer’s Disease"

_ijms, 2024, doi:10.3390/ijms25094639_

Round 1
Reviewer 1 Report
Comments and Suggestions for Authors
The manuscript by Pszczolowska et al., about Chronic Traumatic Encephalopathy (CTE) and Alzheimer’s Disease (AD) reviews the similarities of these two pathologies and discusses current and novel therapeutic approaches for CTE and AD. Although this is a topic of importance to the field, there are a few aspects that need the author's attention before manuscript publication.
- Line 39. It is not clear what the word “confusion” refers to when describing brain pathology in CA1. “Confusion” may not be the right terminology when describing the brain histopathological signatures in AD and CTE, and as such it should be corrected.
- Line 67 and 68. Authors should be more specific as to what is the cell type that accumulates tau protein, as it is very well-known that the intraneuronal accumulation of phospho tau proteins is a hallmark characteristic of AD and/or CTE compared to other neurodegenerative disorders, in which there is rather astrocytic or oligodendrocyte tau accumulation. A sentence like “AD pathogenesis is based on the accumulation of extracellular Abeta and intraneuronal hyperphosphorylated tau proteins” would be more appropriate.
- Lines 97-99 would benefit from a more comprehensive discussion on the current literature concerning Abeta-targeting monoclonal antibodies for AD. A thorough review should encompass the latest advancements and insights in the field, making it imperative to include this crucial information in the present manuscript.
- There are several conceptual errors that require correction in section 4, particularly from lines 184-201. For instance, in line 184, authors mention that “the assembly of Tau into filaments can be initiated or accelerated by adding granules”, but it lacks specificity regarding what "granules" refers to. Are these granules indicative of misfolded tau proteins?. Regarding the statement “expressed tau can only be buried if it is capable of aggregation it is misleading, as tau in healthy individuals is expressed without forming abnormal aggregates. The sentence “Aggregation inhibitors may be responsible for reducing tau-induced seeding and spreading. Tau induces them.” is unclear and could be rephrased to avoid misleading the reader. The statement "Many studies believe that monomeric Tau cannot form aggregates" (line to be specified) is inaccurate according to current literature (Mirbaha et al., 2018). Revisiting the literature is necessary to correct this statement. Moreover, the last paragraph in section 4 appears incomplete. Adding a concluding sentence would effectively wrap up this section.
- There is another conceptual error in line 209 “APP is degraded …. leading to the generation of insoluble Ab, which aggregates into plaques”. This statement oversimplifies the process, as not all Abeta is insoluble nor does it all aggregate into plaques. Instead, various isoforms of Abeta, such as Abeta40 and Abeta42, exist, with Abeta42 being more prone to aggregation and more abundant in AD compared to healthy individuals. This sentence should be revised to accurately reflect these complexities. Additionally, appropriate references should be cited to support these assertions.
- Lines 454-460. Discuss the role of stem cells in the release of bioactive molecules to create a neuroprotective environment, as this is one of the main potential therapeutic strategies that are currently being investigated for the treatment of neurodegenerative diseases.
- Line 499 seems to contain a typo : “therapy is. a set of” . Please, correct.
Comments on the Quality of English LanguageMinor editing of English language required
Author Response
Dear Reviewer,
Thank you for your time and valuable comments. Me and my teammates have corrected our manuscript according to all your suggestions.
We replaced the word 'confusion' and try to explain histopathological changes in better way.
We written more specifically about tau protein, tried to be more specific about it, according to your suggestions.
We have added new sentences in order to explain better Abeta-targeting monoclonal antibodies for AD.
We have carefully reviewed and made many changes in paragraph 4 in order to obtain more clear message. We have changed the world 'granules' and used more specific way describing those features. All the mentioned in the review sentences were rewritten.
This sentence should be revised to accurately reflect these complexities. Additionally, appropriate references should be cited to support these assertions.
We have improved the sentence in line 209. We have also corrected the citation.
We have added new information according to the new sources about stem cells use in therapy.
The mistake in line 499 was corrected
Minor editing of English language was done.
We are open to any further corrections.
Yours sincerely
Magdalena Pszczołowska
Reviewer 2 Report
Comments and Suggestions for Authors
The manuscript attempts to link Chronic Traumatic Encephalopathy (CTE) and Alzheimer's Disease (AD) through shared neuropathological features such as tau protein abnormalities and amyloid-beta (Aβ) accumulation. The paper discusses various epidemiological studies and neuroimaging biomarkers. Below, I direct my suggestions and inquiries to the authors for further clarification and enhancement of their manuscript.
Major Comments
1- The manuscript provides an overview of its literature search parameters; however, it does not specify the nature of the review, whether systematic or narrative. This distinction is critical for understanding the reproducibility and breadth of the findings presented. Could the authors clarify if the review protocol adhered to a specific guideline, such as PRISMA, or another established framework? Furthermore, details regarding the article selection process, including measures taken to minimize selection bias and the criteria for inclusion and exclusion, are notably absent. Incorporating a methodology section with a figure illustrating the search and selection process (initial number of articles and number excluded at each stage) would greatly enhance the clarity and rigor of the review.
2- The methodology section does not adequately describe the control of variables and consideration of confounding factors, which are essential for the accurate interpretation of the study’s findings, particularly concerning biochemical markers in Alzheimer's Disease. Could the authors elaborate on how they addressed potential confounders related to age and disease severity in their analysis?
3- The manuscript suggests a direct correlation between an increase in tau-protein tangles and cognitive decline, a conclusion that overlooks recent studies with contradictory findings. How do the authors reconcile their interpretation with recent research indicating no significant correlation between tau-protein accumulation and cognitive decline in a larger cohort?
4- The manuscript would benefit from improved organization and clearer definitions of key concepts. The transition from discussions on neurobiological mechanisms to therapeutic possibilities, for example, is abrupt. Enhancing the manuscript’s structure with subheadings or a more logical flow could significantly improve readability and comprehension.
5- The discussion section would be enriched by a more critical examination of studies that may contradict or not support the hypothesis presented. Exploring alternative explanations for the observed links between CTE and AD could provide a more balanced perspective and highlight potential avenues for future research.
Minor Comments
1- Please review the citation styles used throughout the manuscript to ensure consistency and clarity, as discrepancies can disrupt the readability and potentially confuse readers.
2- Figures are not referenced in the text.
3- It is crucial to ensure that all cited studies are accurately represented. Misrepresentation of study findings, whether intentional or not, can mislead readers. The authors should double-check that their interpretations of cited research faithfully reflect the original sources.
Author Response
Dear Reviewer,
Thank you for your time and valuable comments. Me and my teammates have corrected our manuscript according to all your suggestions.
We have clarified how we have obtained materials necessary to create the review. We have added paragraph about methodology and resources in the beginning of the manuscript.
We have revised the structure of the manuscript, adding subheadings. We have also rewritten several sentences in order to make our manuscript easier to understand.
We have taken into account recent research indicating no significant correlation between tau-protein accumulation and cognitive decline in a larger cohort in order to your suggestions. We have also enriched the conclusions section.
We have also revised the citations and referred figures in the main text.
We are open to any further suggestions.
Yours sincerely
Magdalena Pszczołowska
Round 2
Reviewer 1 Report
Comments and Suggestions for Authors
Authors have addressed most reviewer's comments, and the manuscript is now suitable for publication
Reviewer 2 Report
Comments and Suggestions for Authors
The revised manuscript has incorporated the suggested changes, and the modifications have been executed as recommended. I endorse this paper for publication.